

# Properties of a cryptic lysyl oxidase from haloarchaeon *Haloterrigena turkmenica*

Nikolay B. Pestov[1], Daniel V. Kalinovsky[1], Tatyana D. Larionova[1],
Alia Z. Zakirova[1], Nikolai N. Modyanov[2], Irina A. Okkelman[1] and
Tatyana V. Korneenko[1]

[1] Shemyakin-Ovchinnikov Institute of Bioorganic Chemistry, Moscow, Russia
[2] Department of Physiology and Pharmacology, University of Toledo, Toledo, OH, United States of America

Corresponding author
Nikolay B. Pestov, korn@mail.ibch.ru

## ABSTRACT

**Background**. Lysyl oxidases (LOX) have been extensively studied in mammals, whereas properties and functions of recently found homologues in prokaryotic genomes remain enigmatic.

**Methods**. LOX open reading frame was cloned from *Haloterrigena turkmenica* in an *E. coli* expression vector. Recombinant *Haloterrigena turkmenica* lysyl oxidase (HTU-LOX) proteins were purified using metal affinity chromatography under denaturing conditions followed by refolding. Amine oxidase activity has been measured fluorometrically as hydrogen peroxide release coupled with the oxidation of 10-acetyl-3,7-dihydroxyphenoxazine in the presence of horseradish peroxidase. Rabbit polyclonal antibodies were obtained and used in western blotting.

**Results**. Cultured *H. turkmenica* has no detectable amine oxidase activity. HTU-LOX may be expressed in *E. coli* with a high protein yield. The full-length protein gives no catalytic activity. For this reason, we hypothesized that the hydrophobic N-terminal region may interfere with proper folding and its removal may be beneficial. Indeed, truncated His-tagged HTU-LOX lacking the N-terminal hydrophobic signal peptide purified under denaturing conditions can be successfully refolded into an active enzyme, and a larger N-terminal truncation further increases the amine oxidase activity. Refolding is optimal in the presence of $Cu^{2+}$ at pH 6.2 and is not sensitive to salt. HTU-LOX is sensitive to LOX inhibitor 3-aminopropionitrile. HTU-LOX deaminates usual substrates of mammalian LOX such as lysine-containing polypeptides and polymers. The major difference between HTU-LOX and mammalian LOX is a relaxed substrate specificity of the former. HTU-LOX readily oxidizes various primary amines including such compounds as taurine and glycine, benzylamine being a poor substrate. Of note, HTU-LOX is also active towards several aminoglycoside antibiotics and polymyxin. Western blotting indicates that epitopes for the anti-HTU-LOX polyclonal antibodies coincide with a high molecular weight protein in *H. turkmenica* cells.

**Conclusion**. *H. turkmenica* contains a lysyl oxidase gene that was heterologously expressed yielding an active recombinant enzyme with important biochemical features conserved between all known LOXes, for example, the sensitivity to 3-aminopropionitrile. However, the native function in the host appears to be cryptic.

**Significance**. This is the first report on some properties of a lysyl oxidase from Archaea and an interesting example of evolution of enzymatic properties after hypothetical horizontal transfers between distant taxa.

## INTRODUCTION

Lysyl oxidase is an amine oxidase that is well characterized in mammals. The human genome contains five lysyl oxidase isoforms (LOX and LOXL1-4), all of them possess the highly conserved C-terminal catalytic domain, the N-terminal signal peptide, and the accessory segments in between. The catalytic domain of LOX is unique among other mammalian amine oxidases because of its ability to oxidatively deaminate various amines including $\varepsilon$-amino groups of lysine residues in peptides and proteins. LOX activity initiates cross-link formation between certain proteins, including elastin, collagen, and fibronectin, and this process is important for maturation and remodeling of the extracellular matrix (*Lucero & Kagan, 2006*). Most animal genomes sequenced to date contain from one to five LOX genes; only placozoans, nematodes, and ctenophores seem to lack any LOX genes (*Grau-Bové, Ruiz-Trillo & Rodriguez-Pascual, 2015*). LOX genes are present in a number of fungal genomes, whereas plants are unknown to possess it. LOX genes are also absent from the vast majority of prokaryotic genomes. Therefore, the presence of true homologues of animal LOX in just several species of Eubacteria and Archaea is of significant interest and reflects the unique history of this enzyme, that most parsimoniously can be explained by multiple horizontal transfer events (HGT) (*Grau-Bové, Ruiz-Trillo & Rodriguez-Pascual, 2015*). Among Eubacteria, LOX genes are frequent in Actinomycetes (especially Streptomyces), some Deltaproteobacteria, occasionally—in other eubacteria, and very rarely—among Archaea. This aspect is exciting not only from the phylogenetic point of view, but also because of potential biotechnological applications, i.e., the fact that distantly related enzymes may have useful properties (*Noda-García et al., 2013*).

It is interesting to note that, in contrast to eukaryotic lysyl oxidases, several LOX homologues identified in prokaryotes exhibit a simple architecture even without a signal peptide (*Grau-Bové, Ruiz-Trillo & Rodriguez-Pascual, 2015*). On the other hand, some prokaryotic LOXes are more complex. Specifically, LOX from *Sorangium cellulosum* (WP_012233967.1) possesses a unique Cys-rich C-terminal non-catalytic domain, which is presumably highly disulfide cross-linked.

The few lysyl oxidase homologues from Archaea that have been sequenced are clustered in two independent groups. This suggests that the two major phyla, Thaumarchaeotes and Euryarchaeotes, may have acquired LOX genes in two independent HGT events (*Grau-Bové, Ruiz-Trillo & Rodriguez-Pascual, 2015*). Indeed, HGT is widespread in Archaea (*Papke et al., 2004*; *Papke et al., 2015*).

*Haloterrigena turkmenica* was isolated from Turkmenistani sulfate saline soil by Zvyagintseva and Tarasov and described in 1987 as *Halococcus turkmenicus* (*Zvyagintseva & Tarasov, 1987*). In 1999, it was proposed to rename it to *Haloterrigena* (*Ventosa et al., 1999*). *H. turkmenica* belongs to the family Halobacteriaceae typus Euryarchaeota and is a fairly fast growing chemoorganotrophic extreme halophile that requires at least 2 M NaCl

with optimal temperature around 45 °C. The complete genome of this archaeon has been sequenced. It consists of 5,440 kbp (including plasmid 6), and was annotated as encoding 5,287 proteins and 63 ncRNAs (*Saunders et al., 2010*).

Here, we attempted for the first time a study on the properties of lysyl oxidase from this haloarchaeon.

## EXPERIMENTAL PROCEDURES

### Materials and strains

A fresh stock of *Haloterrigena turkmenica VKMB-1734* was purchased from the All-Russian Collection of Microorganisms (G.K. Skryabin Institute of Biochemistry and Physiology of Microorganisms, Pushchino, Moscow Region, Russia). Capreomycin was from S.P. Incomed (Moscow, Russia), amikacin from OAO Sintez (Kurgan, Russia), substance P was custom synthesized at Syneuro (Moscow, Russia), hexylamine and 3-aminopropionitrile fumarate from Alfa Aesar (USA), all other amine substrates were from Sigma (USA).

### Cultivation of *H. turkmenica*

Various haloarchaeal media with NaCl around 200 g/l such as INMI medium-3, DSMZ-372 are suitable. Care should be taken to adjust pH since *H. turkmenica* does not grow in acidic media. We found that a simpler medium (hereafter referred to as **IAO**) is a better choice: casamino acids, 5 g/l; yeast extract, 5 g/l; NaCl, 220 g/l; pH 7.6—autoclaved and supplemented with $MgSO_4$, 5 mM; $CuCl_2$, 10 μM. Solid IAO medium may be used for growing single colonies, however, only with high quality agar (some batches inhibit growth). Also, *H. turkmenica* can be easily adapted to a defined medium, hereafter referred to as **MHTU,** an enriched version of HMM (*Mosin & Ignatov, 2014*): L-alanine, 0.4 g/l; L-arginine, 0.4 g/l; D-asparagine, 0.2 g/l; L-aspartic acid, 0.4 g/l; L-cysteine, 0.1 g/l; L-glutamic acid, 1.5 g/l; L-histidine, 0.7 g/l; L-isoleucine, 0.5 g/l; L-leucine, 0.8 g/l; D,L-lysine, 2 g/l; D,L-methionine, 0.4 g/l; L-phenylalanine, 0.3 g/l; L-proline, 0.4 g/l; D,L-serine, 0.6 g/l; L-threonine, 1 g/l; L-tyrosine, 0.2 g/l; D,.L-tryptophan, 0.5 g/l; L-valine, 1 g/l; AMP, 0.1 g/l; NaCl, 220 g/l; $MgSO_4 \cdot 7H_2O$, 20 g/l; KCl, 2 g/l; $NH_4Cl$, 0.5 g/l; $KNO_3$, 0.1 g/l; $KH_2PO_4$, 0.1 g/l; $K_2HPO_4$, 0.1 g/l; $Na_3 \cdot$citrate, 0.8 g/l; $MnSO_4 \cdot 2H_2O$, 0.0003 g/l; $CaCl_2 \cdot 6H_2O$, 0.1 g/l; $ZnSO_4 \cdot 7H_2O$, 0.05 mg/l; $FeSO_4 \cdot 7H_2O$, 0.05 g/l; $CuCl_2$, 10 μM; glycerol, 1 g/l; D-leucine-OH, 0.1 g/l; norleucine, 0.1 g/l; thymine, 0.1 g/l; uracil, 0.1 g/l; pH 7.5.

### Isolation of *Halorubrum sp.* VKK1262

Commercial salt from Upper-Kama deposit (Uralmedprom, Berezniki, Russia) was mixed with IAO medium without NaCl (200 g/l) and filtered through a 0.2 μm GSWP filter (Millipore, USA). Filters were incubated on IAO medium plates prepared with Noble agar (Difco, USA) at 37 °C for one week and colored colonies were restreaked several times on fresh plates. 16S RNA sequence was analyzed by PCR with primers Arch16S-f2 and Arch16S-r934 and Sanger sequencing of the amplicons in both directions.

### Gene cloning

The DNA used as a template for PCR was isolated from the cell culture using a ZR Fungal / Bacterial DNA MicroPrep kit (Zymo Research, USA) according to the manufacturer's

instructions. For PCR, in a total volume of 25 µl, primers (sequences in Supplement) at a concentration of 0.8 µM, PCR buffer 5x Phusion GC reaction buffer, 2 µl 2.5 mM deoxyribonucleotide solution, 0.2 µl of Phusion DNA polymerase, and *H. turkmenica* genomic DNA as a template were used. The cycling parameters were as follows: 1. Hot start 98 °C for 2 min; 2. Denaturation at 98 °C, 30 s; 3. Annealing at 55 °C, 1 min; 4. Elongation at 72 °C, 2 min. 30 cycles between steps 4 and 2. 5. Final elongation at 72 °C for 7 min. Purified polynucleotide fragments HTU-AA and HTU-QV (AA and QV stand for corresponding N-terminal dipeptides in the HTU-LOX sequence) were digested with *Bam* H I and *Hin* d III restriction enzymes and ligated into the corresponding sites of the pQE-30 vector (Qiagen, USA), followed by transformation of the *E. coli* strain XL1-Blue by electroporation. Colonies screening was performed by PCR, and the sequence was confirmed by Sanger sequencing.

## Protein expression

The XL-1 Blue transformants HTU-AA and HTU-QV were grown in LB medium containing ampicillin on an orbital shaker at 37 °C until $OD_{600} = 0.7$ was reached, followed by induction of expression with 0.5 mM IPTG for 3 h. The cells were then harvested by centrifugation and stored at −70 °C. His-tagged proteins were purified under denaturing conditions (8 M urea) on the metal-chelating sorbent Ni-NTA agarose (*Korneenko et al., 1997*). Typical yields of the purified proteins HTU-QV and HTU-AA were around 25–27 mg per liter of culture. The resulting proteins in 8 M urea pH 6.3 buffered with 0.5 M imidazole, 0.1 M sodium phosphate, and 20 mM Tris were dialyzed against different buffers (optimization briefly described in Results).

## Activity assays

Determination of substrate specificity was performed using a fluorometric method suitable for various amine oxidases as the release of hydrogen peroxide coupled to the oxidation of 10-acetyl-3,7-dihydroxyphenoxazine (Biotium, Germany), also known as Amplex red, in the presence of horseradish peroxidase (*Palamakumbura & Trackman, 2002*). The fluorescence of the reaction product (resorufin) was assayed with a Microplate analyzer "Fusion" (Perkin Elmer, USA) at excitation and emission of 535 and 620 nm, respectively. More specifically, the reaction was carried out in 0.1 M borate buffer pH 8.3 in the presence of 1 U/ml horseradish peroxidase at 37 °C, 100 µl volume and 0.5–5 µg purified protein (up to 50 µg protein for crude lysates). For the negative control, 0.1 mM 3-aminopropionitrile was added to block any lysyl oxidase activity. Calibration has been done with known amount of hydrogen peroxide, and data were fitted to Michaelis–Menten equation using Prism software package (GraphPad, USA). Sheep LOX was isolated from aorta as described before for the purpose of comparison with HTU-LOX (*Pestov et al., 2011*).

*Immunization* of rabbits was carried out with purified folded protein HTU-QV. Initially, rabbits were subcutaneously injected with 100 µg protein as an emulsion in Freund's complete adjuvant. The first booster injection was made with the same quantity of the antigen in incomplete Freund's adjuvant 5 weeks after the first immunization, and the second booster injection—with 250 µg antigen and no adjuvants 6 weeks later. One week

after the second booster injection, sera were collected and stored with the preservative sodium azide at 4 °C. Immunization of rabbits has been approved by Animal Care and Use Review Board of Shemyakin-Ovchinnikov Institute of Bioorganic Chemistry, protocol No 15/2011.

*Affinity purification of antibodies* was performed on a small-scale essentially as before (*Pestov et al., 2004*) using purified HTU-QV protein electrophoresized using SDS-PAGE and blotted on a PVDF membrane: the HTU-QV band on PVDF was blocked in TBST buffer containing 5% bovine serum albumin and 5% non-fat milk, then serum was added and incubated for 3 h. After several washes with TBST bound antibodies were eluted with 0.1 M sodium citrate (pH 2.0) for 10 min, followed by immediate neutralization with unbuffered Tris and addition of 0.1% bovine serum albumin and 0.02% sodium azide.

## Western blotting

*H. turkmenica* cells were centrifuged, and the pellets were lyzed in 10 mM Tris-HCl, 1 mM $MgCl_2$, pH 7.5 containing 0.5 mM tris(2-carboxyethyl)phosphine (Sigma, USA), Complete Protease Inhibitor Cocktail (Roche, Switzerland), and 1 u/ml Benzonase (EMD Millipore, USA) for 15 min at 37 °C, followed by centrifugation for 10 min at 15,000 g. The supernatants were mixed with Laemmli sample loading buffer without mercaptoethanol and analyzed by electrophoresis in 8% SDS PAGE gels. Following electrophoresis, the protein samples and colored protein weight markers (Spectra Multicolor Broad Range, Thermo, USA) were transferred from polyacrylamide gel onto a PVDF membrane (GE Healthcare, USA). The membrane was washed for 5 min with 2% SDS, then blocked in TBST buffer containing 5% non-fat milk, 0.02% sodium azide, and 10% w/w Bløk blocker (EMD Millipore, USA) overnight at 4 °C. On the next stage, the membrane was incubated in 10 ml of TBST buffer solution with 0.1% non-fat dry milk and primary rabbit antibodies (1:10,000) for one hour at room temperature, rinsed out with TBST buffer 10 times for 5 min each, followed by incubation with secondary antibodies (HRP-conjugated anti-rabbit antibodies, Biotium, Germany, 1:50,000) in 10 ml TBST buffer with 0.1% non-fat dry milk for one hour, and rinsed out again in the same way. Chemiluminescence was recorded using Femto Maximum Sensitivity Western Blotting Detection Reagent (Thermo, USA) and Carestream Kodak Biomax Light film (Sigma, USA).

## RESULTS

We initially attempted to produce the full-length HTU-LOX protein in *E. coli* but found that it precipitates as inclusion bodies without any detectable amine oxidase activity (Supplemental Information), and all attempts at its refolding were unsuccessful (results not shown). For this reason, we proceeded to truncations without the N-terminal peptide (hydrophobic segments are common sources of problematic expression in *E. coli*) with subsequent purification under denaturing conditions and refolding. The purity of the resulting eluate was checked by SDS PAGE. Figure 1 illustrates the expression and purification of HTU-LOX exemplified by HTU-QV variant. Of note is the fact of its anomalously slow electrophoretic mobility that corresponds to an apparent molecular weight of 34 kDa, whereas the theoretical value of the His-tagged HTU-QV is 24.3 kDa.
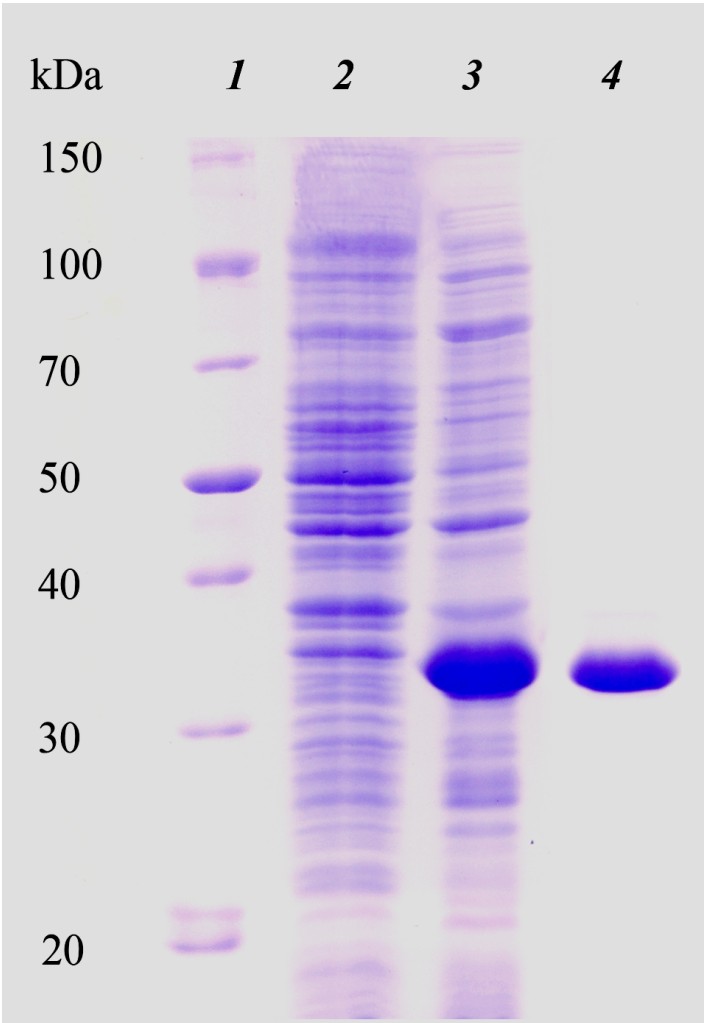

**Figure 1** **Electrophoretic analysis of recombinant N-terminally truncated *H. turkmenica* LOX (HTU-QV) expression and purification.** 1, molecular weight marker proteins; 2, *E. coli* proteins before addition of IPTG; 3, expression induced with IPTG; 4, purified HTU-QV protein.

Since HTU-LOX is a rather acidic protein (theoretical pI 4.58 for the His-tagged HTU-QV and 4.09 for the full-length protein), this peculiarity is common among acidic proteins (*García-Ortega et al., 2005*).

Refolding of the purified His-tagged proteins HTU-AA (Δ1-34) and HTU-QV (Δ1-91) was achieved using dialysis against different buffers and results in good amine oxidase activity. We investigated a variety of factors that may improve the formation of catalytically active proteins HTU-AA and HTU-QV: buffer type and concentration (Tris, phosphate buffered saline, acetate, etc.), the ionic strength of the solution (concentration of NaCl), temperature, metal ions (Cu, Fe, Zn, Ni, Co, Mn) in different concentrations, pH of the solution (5.0–8.0), as well as dialysis with a gradual decrease in the concentration of the denaturing agent (urea). Optimal pH is around 6.2 (Fig. 2A). Since it is known that

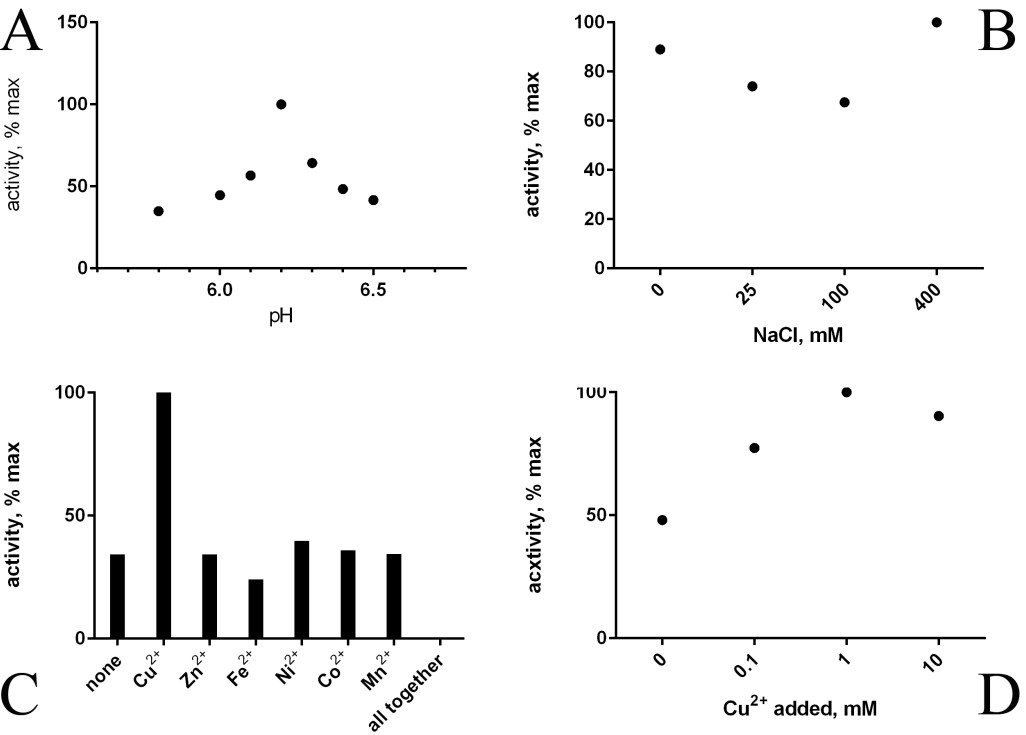

**Figure 2 Folding of recombinant *H. turkmenica* lysyl oxidase (HTU-QV).** Folding by dialysis. (A) Influence of pH; (B) effect of NaCl; (C) 1 mM salts of various metals added to the dialysis buffer; (D) different concentrations of CuSO$_4$.

mammalian LOX requires the presence of a copper ion in the catalytic domain in order to achieve the formation of the lysyl-tyrosine quinone (LTQ) in the catalytic center, we expected similar results for HTU-LOX. Indeed, only Cu$^{2+}$ increases activity (Figs. 2C and 2D), whereas a mixture of different ions gives an inhibition (Fig. 2C). It is interesting to note that refolding efficiency is only slightly affected by NaCl concentration, contrary to the expectations from the fact that *H. turkmenica* is an extreme halophile that requires at least 2 M NaCl (Fig. 2B). Similar results were obtained in folding by dilution experiments, demonstrating also that 1 M and 2 M NaCl cannot improve activity any further (results not shown).

A slow decrease of the denaturant (urea) concentration was found to lack any advantage over the stepwise approach with immediate transfer into a buffer without urea. This was confirmed by refolding by dilution (results not shown). Ultimately, a simple refolding procedure may be chosen as optimal:

- Dialysis against 40 mM sodium acetate, pH 6.2 with 1 mM CuSO$_4$ at 4 °C for 3 h;
- Dialysis against 40 mM sodium acetate, pH 6.2 without copper at 4 °C overnight.

Interestingly, under any conditions used, the amine oxidase activity of the protein HTU-QV (typical activity for HTU-QV with 1 mM taurine at pH 8.3 was approximately 0.014 μmole/min hydrogen peroxide per mg protein) was about fifteen times higher than

**Table 1  Substrate efficiencies of recombinant *Haloterrigena* lysyl oxidase (HTU-QV variant) in comparison with mammalian enzymes from aorta.** $V_{max}/K_m$ ratios normalized with respect to tyramine. Values for polymers and lysozyme calculated as for molar amine groups. Data for LOX from bovine aorta from *Shah et al., (1993)* (note that conditions used were significantly different from this study). For HTU-LOX the short variant QV has been used.

| Substrate | HTU-LOX | Sheep | Bovine |
|---|---|---|---|
| L-lysine | 0.058 | 0.028 | ND |
| cadaverine | 0.370 | 1.070 | 1.09 |
| histamine | 0.550 | 0.920 | ND |
| taurine | 1.120 | 0.120 | ND |
| glycine | 0.020 | z | ND |
| $\beta$-alanine | 0.005 | z | ND |
| GABA | 0.015 | z | ND |
| methylamine | 0.020 | ND | ND |
| substance P | 0.068 | ND | ND |
| lysozyme | CK | 0.080 | ND |
| polyallylamine | CK | 0.080 | ND |
| amikacin | 0.260 | ND | ND |
| capreomycin | 0.120 | 0.190 | ND |
| polymyxin | 0.780 | ND | ND |
| benzylamine | z | 0.170 | 0.52 |
| hexylamine | 0.28 | 1.140 | 0.14 |

**Notes.**
ND, no data; Z, rate too low for accurate determination; CK, complex kinetics with inhibition by substrate at high concentrations.

that of HTU-AA. Therefore, the segment of HTU-LOX sequence from Ala$^{35}$ to Gln$^{92}$ may function as an inhibitory (pro)peptide or just interferes during refolding.

Refolded proteins HTU-QV and HTU-AA exhibit activity against a wide variety of primary amines (Table 1): histamine, methylamine, lysine, cadaverine, tyramine, etc. Even glycine and $\beta$-alanine are efficiently oxidized, in contrast with mammalian LOX. HTU-LOX readily oxidizes some amine-containing antibiotics: polymyxin and aminoglycosides such as capreomycin and amikacin. This is a unique property of lysyl oxidases, since other amine oxidases either do not deaminate aminoglycosides or are even inhibited by them, as in the case of *E. coli* amine oxidase (*Elovaara et al., 2015*). Regarding various proteins, lysine-containing peptides, and polymers (e.g., poly-L-lysine, poly-allylamine, lysozyme, and substance P as an example of a Lys-containing peptide), HTU-LOX behaves almost like LOX from the aorta. Taurine is one of the best substrates for HTU-LOX. It is also capable of oxidizing glycine, $\beta$-alanine, and $\gamma$-aminobutyric acid. The only amine that HTU-LOX oxidizes much worse than mammalian LOX is benzylamine. Importantly, HTU-LOX demonstrated good sensitivity to the classical inhibitor of all LOXes—3-aminopropionitrile (BAPN). Also, HTU-LOX is somewhat different from the mammalian enzyme in terms of pH dependence. In contrast to the latter, HTU-LOX activity does not exhibit a steep decline from its maximum around 8.3, and even displays a certain degree of bimodality retaining some activity even below 7 (Fig. 3).

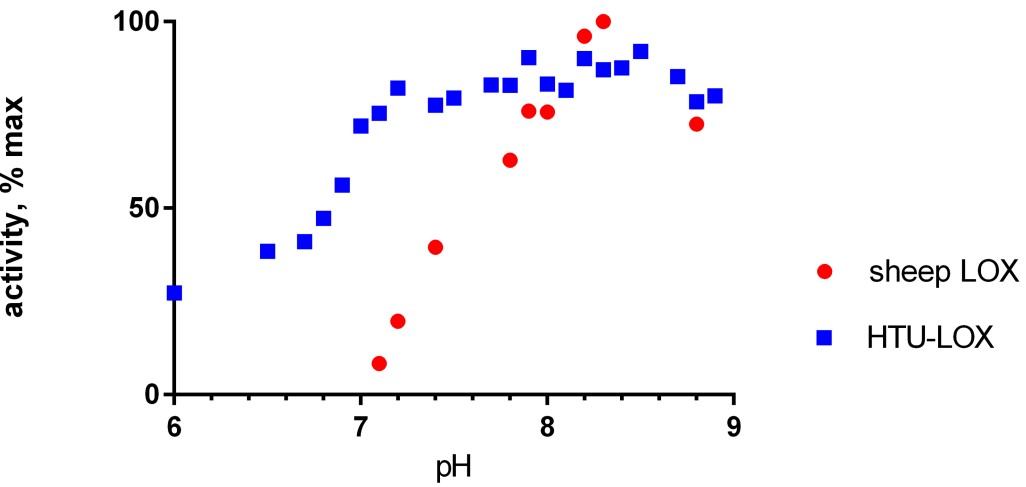

**Figure 3** **pH dependence of recombinant *H. turkmenica* lysyl oxidase.** In comparison with LOX from sheep aorta amine oxidase reaction rates were measured for HTU-QV protein if universal borate-phosphate-acetate buffer with histamine as the substrate.

We also attempted to study the HTU-LOX protein in the host—the archaeal halophile *H. turkmenica*. For this purpose, we raised polyclonal antibodies against the truncated HTU-LOX (variant QV). The full-size HTU-LOX theoretically contains 308 amino acids with a molecular weight of 33,829 Da, whereas the full-length protein expressed in *E. coli* has electrophoretic mobility corresponding to 52 kDa (Fig. S1). However, this apparently large discrepancy should be regarded as normal, since the anomalous mobility has been observed for purified recombinant HTU-LOX (Fig. 1). Western blotting indicates that HTU-LOX may be present in *H. turkmenica* cells (Fig. 4, lanes 1–2; compare to negative controls in lanes 3–6, where lanes 4–6 show non-specific binding of unrelated antibodies plus secondary antibodies, and lanes 3 and 6 provide an additional negative control—non-specific binding of antibodies to proteins from *Halorubrum,* a totally different haloarchaeal species). It should be emphasized that specific bands were detected only at a high sensitivity, meaning that the normal expression level of the protein in cultured *H. turkmenica* is quite low, and detection of the full-length, unprocessed HTU-LOX was obscured by non-specific bands (Fig. 4, lanes 4–5). Most interestingly, only a very high molecular weight band specific to anti-HTU-LOX antibodies of about 210 kDa has been reliably detected. Importantly, this band is certainly absent from the *Halorubrum* sample (Fig. 4, lane 3), where any HTU-LOX should be absent *a priori*. Prolonged incubation of the cells in saturated salt in the medium results in a marked decrease in the intensity of this band (Fig. S1). Therefore, we can hypothesize that HTU-LOX in *H. turkmenica* predominantly exists in a moderately stable homo- or heterooligomeric form, and its migration in SDS-PAGE corresponds to an apparent molecular weight of 210 kDa, however, this anomaly awaits further studies.

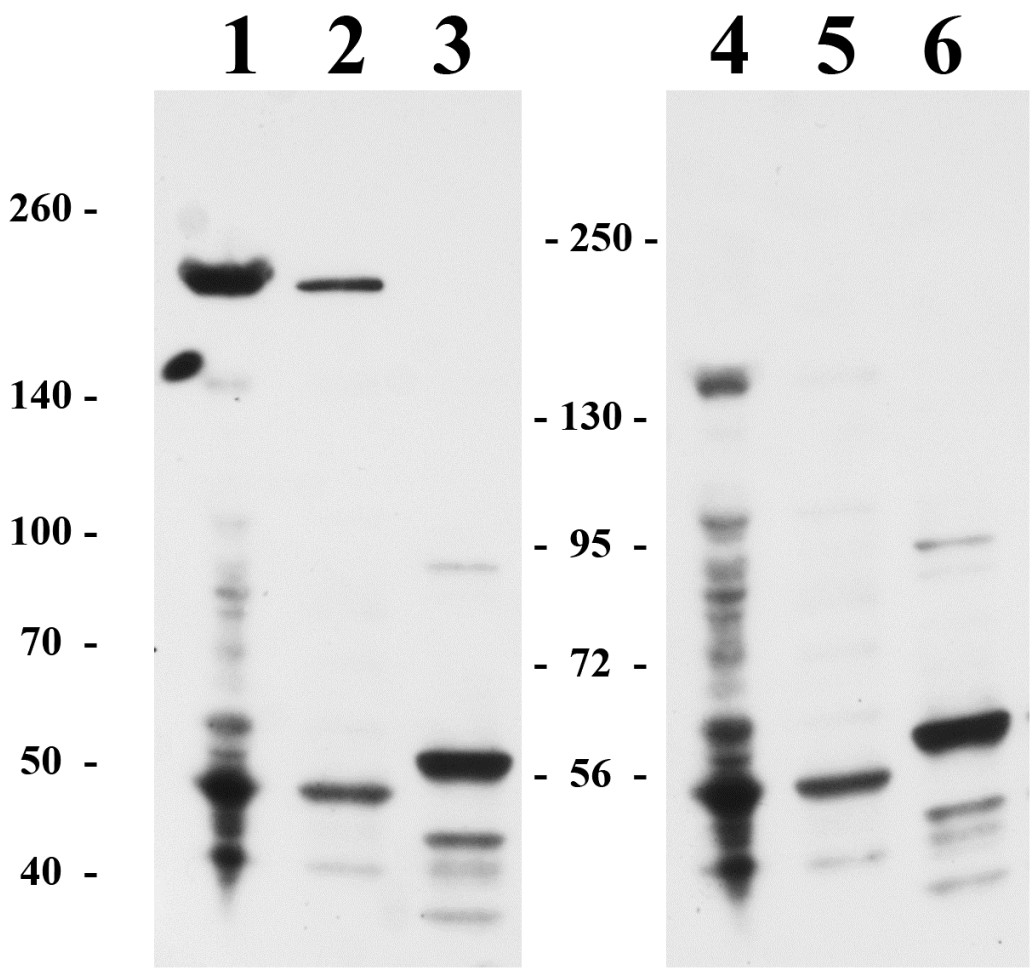

**Figure 4  Immunoblotting detection of lysyl oxidase in *H. turkmenica* cells with anti-HTU-LOX anti-bodies.** Chemiluminescence of bound HRP-labeled antibodies; positions of molecular weight markers on the left and in the middle. 1–3, affinity purified antibodies against HTU-QV, the catalytic domain of LOX from *H. turkmenica*. 4–6, negative control with rabbit serum against an unrelated antigen (*Dmitriev et al., 2009*). 1, 2, 4, 5, proteins of *H. turkmenica* grown to log-phase (lanes 1 and 4 were loaded with three times more protein than lanes 2 and 5). 3, 6, proteins of *Halorubrum sp. VKK1262* (loading equal to lanes 2 and 5).

We also found that BAPN (even at a rather high concentration of 1 mM) had no significant effect on sensitivity of fresh cells to osmotic stress, on formation of hypotonically-resistant cysts, or on growth rate in both conventional (IAO) and defined (MHTU) media.

## DISCUSSION

Amino acid sequence alignments (Figs. 5 and 6) of LOX proteins demonstrate poor overall conservation (for example, high variability in the number of disulfide bonds) with only a few hyperconserved amino acid residues like Cu-binding His and LTQ formation (*Zhang et al., 2018*). A fundamental aspect that needs to be emphasized is the fact that relatively

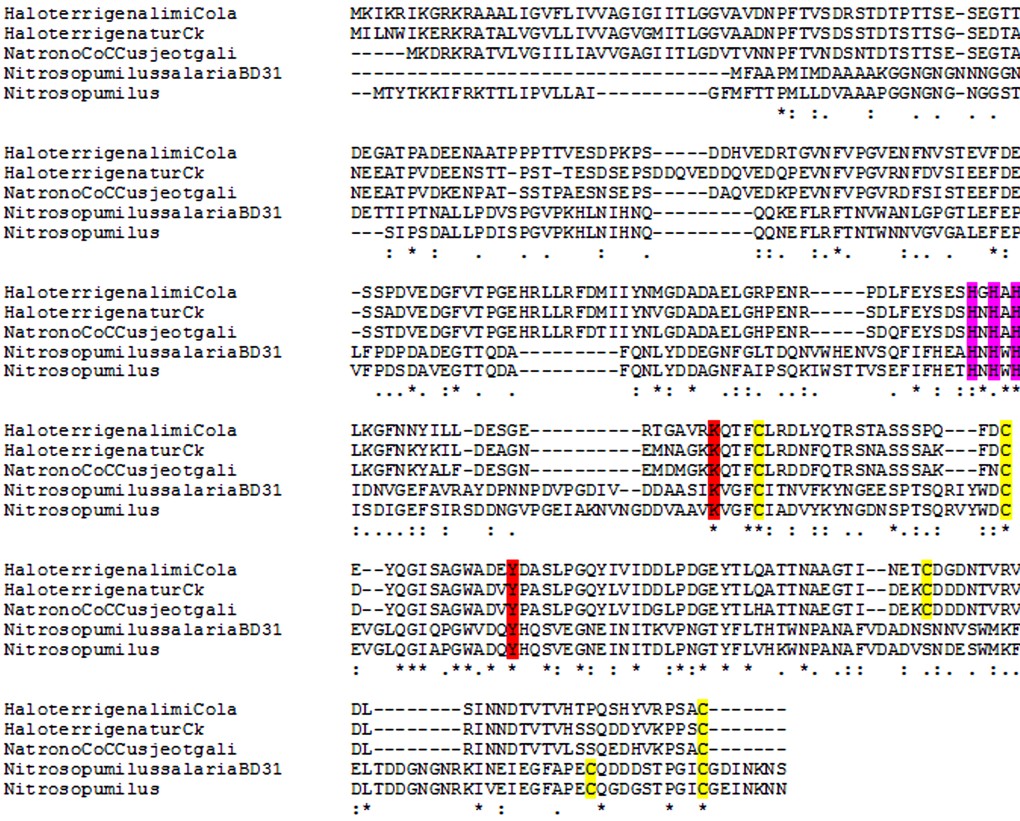

**Figure 5** **Multiple alignment of archaeal lysyl oxidases.** Obtained using Muscle algorithm. (Haloterrigenalimi­Cola, *Haloterrigena limicola*; HaloterrigenaturCk, *Haloterrigena turkmenica*; NatronoCoCCusjeot­gali, *Natronococcus jeotgali*; NitrosopumilussalariaDB31 and Nitrosopimulus, *Nitrosopumilus sequnces*). Yellow, cysteine residues marked in yellow; red, LTQ-forming lysine and tyrosine; purple, three hypercon­served histidine residues necessary for the binding of $Cu^{2+}$.

little research has been carried out on the influence of HGT with the subsequent adaptation of the catalytic properties of the enzymes to a new host.

Refolding efficiency is not significantly affected by NaCl concentration. This surprising fact could reflect the history of prokaryote LOX genes: halophile archaea may have acquired these genes from microorganisms with a rather different requirement for salt. The ancient HGT event may had even originated from a halophobic organism, followed by "domestication" that suppressed the formation of misfolded protein. Besides, LOX may had served as an antibiotic resistance enzyme under aerobic conditions. This, however, is unlikely in extant *H. turkmenica*, since Archaea are usually highly resistant to both polymyxin and common aminoglycosides. Also, HTU-LOX oxidizes some peptide antibiotics and theoretically this feature may be useful for competition with other species of haloarchaea (*Besse et al., 2015*) in the natural habitat of *H. turkmenica*. The low expression level of the enzyme suggests that HTU-LOX plays a modest functional role in increasing availability of nitrogen from non-typical amines. Its promiscuous substrate specificity and

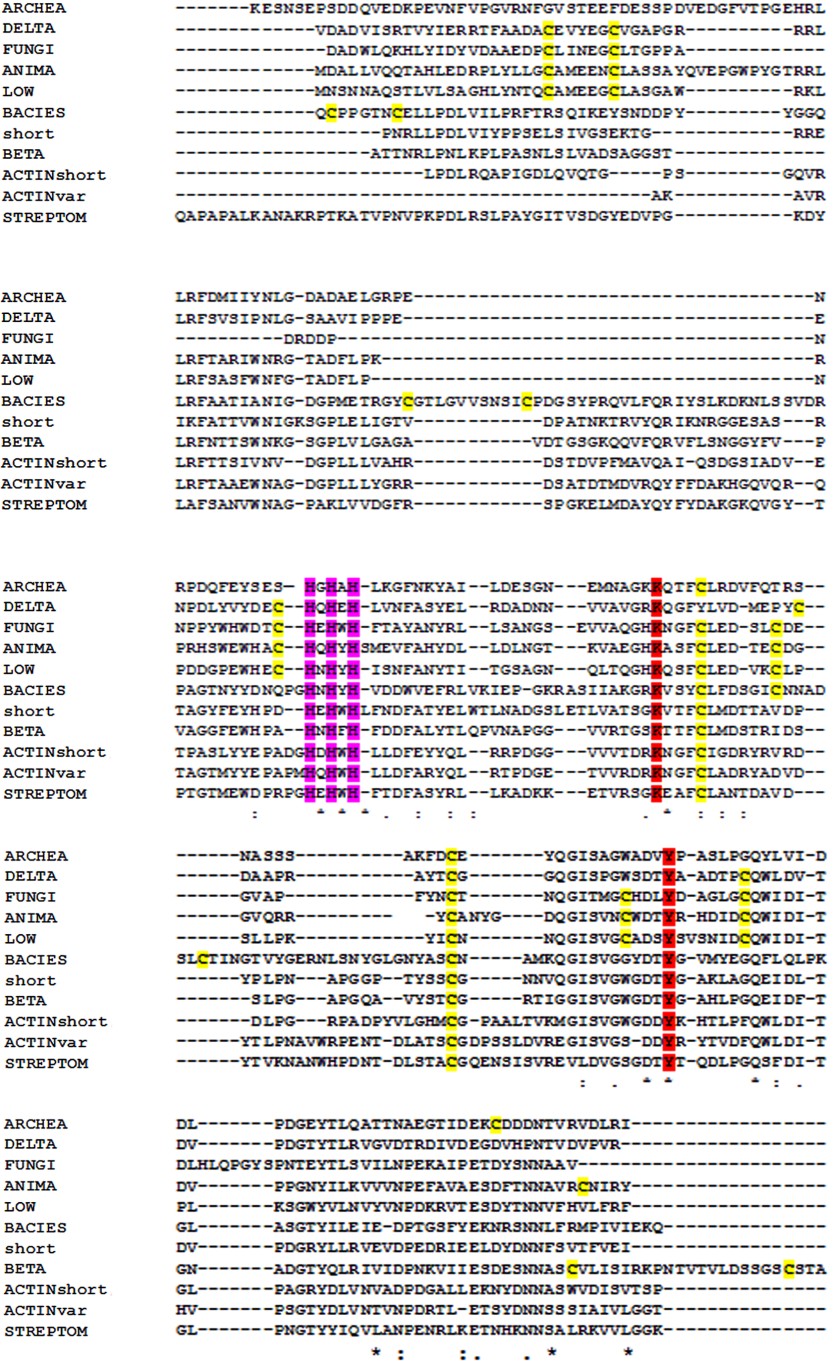

**Figure 6 Multiple alignment of the conserved segments of catalytic domains from all lysyl oxidases representing different kingdoms.** Obtained using Muscle algorithm from consensus sequences of different taxa. ARCHAE, Archaeal LOX sequences; DELTA, *Deltaproteobacteria*; FUNGI, fungal LOXes; ANIMA, various animal LOXes; LOW, *Mesomycetozoa* and *Orthonectida*; short, LOX from *Truepera radiovitrix, Deinococcus pimensis, Nitrospira nitrosa,* and a few samples from Parcubacteria; BETA, *Betaproteobacteria*; Actinshort, *Amycolatopsis mediterranei* LOX and its closest homologues; STREPTOM, LOX from *Streptomyces*; ACTINvar, other actinomycetal LOXes; BACIES, all other eubacterial LOXes. Yellow, cysteine residues marked in yellow; red, LTQ-forming lysine and tyrosine; purple, three hyperconserved histidine residues necessary for the binding of copper.

negligible enzymatic activity in *H. turkmenica* cells make it difficult to demonstrate this fact experimentally.

HTU-LOX accepts glycine, $\beta$-alanine, and $\gamma$-aminobutyric acid as substrates. This observation is unusual, because the presence of any acidic groups in vicinity of the amino group almost completely prevents oxidation by most amine oxidases. Thus, it is safe to conclude that HTU-LOX has a relaxed substrate specificity in comparison with its mammalian homologue (*Shah et al., 1993*). Perhaps a low selection pressure on the lysyl oxidase gene allowed it to lose substrate specificity. This, however, may be useful for biotechnological purposes as a starting point for molecular evolution in any direction.

Another interesting fact is that the amine oxidase activity of the truncated protein HTU-QV is much higher than that of the longer one, HTU-AA. This observation is in line with the general view that LOX catalytic domain is usually (except for animal LOXL2-4 and homologs) preceded by an autoinhibitory sequence, together forming a propeptide. In the case of HTU-LOX, the autoinhibitory sequence corresponds to the stretch from $Ala^{39}$ to $Gln^{92}$. However, the inhibition is relatively inefficient, and this may also reflect the evolution of HTU-LOX gene after the in-Archaea HGT that resulted in a partial degradation of the autoinhibitory function of the propeptide. LOX genes in Archaea underwent at least two independent HGTs (*Grau-Bové, Ruiz-Trillo & Rodriguez-Pascual, 2015*), and this is just an example of HGT in Archaea (*Papke et al., 2004*; *Papke et al., 2015*). The widespread occurrence of these HGT events may also indicate that the transferred genes not necessarily possess indispensable functions in every species.

Immunoblotting experiments allow us to hypothesize that HTU-LOX in *H. turkmenica* predominantly exists in a moderately stable homo- or heterooligomeric form, which migration in SDS-PAGE corresponds to an apparent molecular weight of 210 kDa. This anomaly awaits further studies but may have interesting implications, for example, explain lack of detectable amine oxidase activity in *Haloterrigena* cells.

What is the origin of the animal lysyl oxidase? Has it emerged in primitive animals at the beginning of their evolution through HGT from Eubacteria? Or, conversely, LOX genes, which have important functions in animals, made their way several times into the world of prokaryotes? The second option seems highly unlikely due to splitting of animal ORFs into exons and molecular phylogeny (*Grau-Bové, Ruiz-Trillo & Rodriguez-Pascual, 2015*) but cannot be excluded completely. In any respect, the most parsimonious explanation of the evolution of the catalytic LOX domain is that inter-kingdom saltations of LOX genes between distant branches of Life occurred more than once.

## CONCLUSIONS

*H. turkmenica* LOX (HTU-LOX) was successfully expressed in *E. coli*. Optimal refolding conditions are different from those for the growth of the host cells. HTU-LOX has a relaxed substrate specificity in comparison with mammalian LOX, benzylamine is a poor substrate for both, and sensitivity to 3-aminopropionitrile is conserved in HTU-LOX. N-terminal truncation of HTU-LOX increases its activity. Cultured *H. turkmenica* does not exhibit any detectable amine oxidase activity, and expression level of the HTU-LOX is low. Therefore, native function of *H. turkmenica* lysyl oxidase may be cryptic.

### Funding
These studies were supported by MCB RAS and Russian Foundation for Basic Research (grant No 16-04-01755), and experiments with antibiotics were supported in part by Russian Science Foundation (grant No 16-14-10335). The funders had no role in study design, data collection and analysis, decision to publish, or preparation of the manuscript.

### Grant Disclosures
The following grant information was disclosed by the authors:
MCB RAS and Russian Foundation for Basic Research: 16-04-01755.
Russian Science Foundation: 16-14-10335.

### Competing Interests
The authors declare there are no competing interests.

### Author Contributions
- Nikolay B. Pestov conceived and designed the experiments, performed the experiments, analyzed the data, contributed reagents/materials/analysis tools, prepared figures and/or tables, authored or reviewed drafts of the paper, approved the final draft.
- Daniel V. Kalinovsky and Tatyana D. Larionova performed the experiments, analyzed the data, prepared figures and/or tables, authored or reviewed drafts of the paper, approved the final draft.
- Alia Z. Zakirova performed the experiments, approved the final draft.
- Nikolai N. Modyanov analyzed the data, contributed reagents/materials/analysis tools, authored or reviewed drafts of the paper, approved the final draft.
- Irina A. Okkelman conceived and designed the experiments, performed the experiments, analyzed the data, authored or reviewed drafts of the paper, approved the final draft.
- Tatyana V. Korneenko conceived and designed the experiments, performed the experiments, analyzed the data, prepared figures and/or tables, authored or reviewed drafts of the paper, approved the final draft.

### Animal Ethics
The following information was supplied relating to ethical approvals (i.e., approving body and any reference numbers):
Immunization of rabbits was approved by the Animal Care and Use Review Board of Shemyakin-Ovchinnikov Institute of Bioorganic Chemisrty, protocol No 15/2011.

### Data Availability
The raw data are available in the Supplemental Files.

### Supplemental Information
Supplemental information for this article can be found online at http://dx.doi.org/10.7717/peerj.6691#supplemental-information.

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
