# Peer review of "Properties of a cryptic lysyl oxidase from haloarchaeon Haloterrigena turkmenica"

_PeerJ, doi:10.7717/peerj.6691_

## Round 0.1 · original submission · Major Revisions

All reviewers highlighted several shortcomings throughout your manuscript. Most salient among these are :

- insufficient description of alignment methods, sequences selected, rationale behind the -AA and -QV constructs

- apparent low specificity of the antibodies used. I concur with reviewer #2 regarding the difficulty of interpreting the faint 50 kDa band LOX. How can you be sure that the intended protein is not present as a hypothetical dimer/trimer with high molecular weight obscured by the very intense non-specific signal with MW between 60 and 100 kDa?

- insufficiently supported speculation

I think a purification table should also be present, as well as a discussion of the inability of generating correctly-folded full length constructs in E. coli: how commonly is such a difficulty observed? Related to this: how sure are you that the gene product even does fold correctly in H. turkmenica ? Since no data on the LOX activity in H. turkmenica extracts are present, I am not sure your data can be used to reliably claim that the (presumably) horizontally gene transferred LOX gene is translated to an active, metabolically significant, protein in H. turkmenica.

Reviewer 1 ·

Basic reporting

. Although this is the first time to study the properties of lysyl oxidase from this archaeon as the authors claimed, the theoretical reasons for many experiment is not provided enough. Also, the manuscript is not prepared properly. Please check the attached file for detail.

Experimental design

The rationale or methods is not provided enough. For example, the authors used two different polymerase but did not explained the reason. The information for the primers are not provided. Also, the rationale of making truncated LOX is not provided. Please check the attached file for detail.

Validity of the findings

Ther are some interesting findings but too many guesses which are not based on the results are inclued such as lines 241, 243, 244 etc.Please check the attached file for detail.

Additional comments

In this articles, the authors reported the expression and functional analysis of LOX protein of haloarchaeon Haloterrigena turkmenica. Although this is the first time to study the properties of lysyl oxidase from this archaeon as the authors claimed, the theoretical reasons for many experiment is not provided enough. Also, the manuscript is not prepared properly. All suggestions and questions are marked on the attached file to save the space and the major concerns or problems are listed below. The authors could submit again after all this issues are resolved.

1. Line 12: the corresponding author is not marked on the list of the authors
2. Line 58-58: the presence of LOX gene some species cannot be the evidence of HGT.
3. The authors used two different polymerase but did not explained the reason. The information for the primers are not provided. Also, the rationale of making truncated LOX is not provided.
4. There are many places that are not proper in description such as Lines 88-89, 91, 93.
5. There are many places at which the location of marked sentence in the attached file is not proper: for example, descriptions that need to be in the discussion section are in the Results section such as lines 125-128, 175-176, 178-180, 200-203, and results are in the discussion section such as lines, 230-232..
6. There are too many guesses which are not based on the results including lines 241, 243, 244 etc.
7. Some explanation is not theoretical: For examples the authors explained the low level expression of HTU-LOX is its moderate functional role in increasing availability of nitrogen from non-typical amines (lines 249-250), which does not accords to its function described on ln lines 53-53.
8. Table 1. The authors said that data for bovine aorta is from other paper at which the reaction condition ist significantly different. In that case the data should not be used for comparison.
9. There are too much unnecessary explanations in the Figure Legends which are marked in the attached file.
10. Figure 5. Is this alignment necessary?
11. Figure 6. The identity of HaloterrigenalimiCola is not provided.
12. Figure 7:. The methods for selecting the sequence as the representative of each domain is not provided.
13. Figure 8. If there is not enough space for full Species name, the GenBank accession number of each need to be provided. Figure 8 shows the relationship among different LOX gene based on the sequence of catalytic domain and it does not show the wat of gene transfer as the authors claimed on lines 232-235

Annotated reviews are not available for download in order to protect the identity of reviewers who chose to remain anonymous.

Reviewer 2 ·

Basic reporting

The manuscript by Pestov et al investigates the properties of the lysyl oxidase form present in the archaeon Haloterrigena turkmenica. While the validity of the findings and experimental design is discussed in the following sections, different aspects referring to the reporting of the manuscript are evaluated here. First, in general, the language used throughout the text is acceptable and structure of the manuscript conforms the journal standard. With respect to whether introduction properly shows the background of the field, there is a particular sentence that does not actually reflect what it is already known on phylogeny of LOX enzymes. Thus, in line 56 it is said that “most fungi are unknown to possess (LOX)”, whereas Grau-Bove et al, 2015 in a very comprehensive analysis of LOX isoforms in a wide variety of organisms already described the presence of homologues of this enzyme in some fungi. Regarding the figures, it is counter-intuitive to see plots composed by a group of dispersed points. My suggestion is to make bargraphs where a non-numerical x axis is displayed (Figure 2C), and line graphs in those cases representing a numerical axis (Figure 2A, B, D and Figure 3). Additionally, in the alignment shown in Figure 5, it is also difficult to see how the different deletion LOX constructs of H. turkmenica were designed and whether this makes sense in the context of the mouse sequence. Finally, information about the sequences used in the alignment shown in Figure 8 is not provided, nor the details of the method employed to generate it.

Experimental design

Original research in the manuscript is within the scope of the journal. Research question is in fact not clear: whether the authors want to analyze the properties of the H. turkmenica form or they want to have a look to the phylogeny of the LOX homologues. The former can be in fact interesting in the context of the mammalian forms, the latter has been already done in detail in the article by Grau-Bove et al, 2015.
With respect of the technical standard of the investigation, as discussed in the next section, protein characterization methodologies are of poor quality. Methods do not include any information about how alignments were done (insufficient description in the figure legend).

Validity of the findings

With respect to the findings, the authors claim that they were able to express, purify and refold a couple of deletion mutants of LOX from H. turkmenica, whereas Fig. 1 only shows the expression of one of them. To be more illustrative with the differences between these proteins and the mammalian homologues, optimization of refolding of these protein forms (Figure 2 only for one construct) should be compared with the mammalian homologue. This will answer the question of whether mouse or human LOXs present the same difficulties for purification, refolding and if they are equivalent in terms of activity. In any case, there is no information about the robustness of the data, whether they are replicates from different preparations, no statistical approach seems to be used.
The authors claim that they are generated a polyclonal antibody against the H. turkmenica protein and used it to characterize the expression of this form in these cells. In fact, quality of the blot is poor, and it is difficult to believe that the faint band observed at about 50 kDa can actually correspond to the endogenous LOX homologue, and therefore, it is very speculative to say that this can be a precursor form. Even more unrealistic is the statement that the growth of the cells without yeast extract resulted in the formation of a band of 3 kDa less mobility.
Finally, to my opinion, multiple alignment and phylogenetic analysis is also very poor, particularly considering the complete study performed in Grau-Bove et al, 2015.

Reviewer 3 ·

Basic reporting

Please, add more details to Materials and Methods section. And all tests should be repeated in triplicate, and the authors should use proper statistical methods for the data in the paper.

Experimental design

no comment.

Validity of the findings

no comment.

Additional comments

Nikolay B Pestov et al., reported the heterologous expression and characterization of a novel lysyl oxidase from haloarchaeon Haloterrigena turkmenica. The author performed some results about the LOX, but the quality of the manuscript is far from publication. Here are some suggestions:

Major points:
1. The “Method” part did not introduce the methods of tests used in this study such as Line 171-174 how to prepare the test of properties with different buffers; Line 190-208 activity measurement of substrates with this enzyme; Fig. 5-7, the aliment of LOX protein sequences; Fig. 8, the reconstruction method of LOX phylogenetic tree.
2. The “Method” part should be more detailed on: 1) “activity assays”, how to define the activity unit should be added, because the activity should have the Unit and should be compared using Unit. 2) the primers used in this study should be summarized in a Table, and the sequences of genes used in this study should be submitted to Genbank or shown in supplementary files. And it is not clear what HTU-AA, HTU-QV and HTU-LOX represents.
3. All tests should be repeated in triplicate, and the authors should use proper statistical methods for the data in the paper.


Minor points:
Line 31, “usual” to “regular”.
Line 38, why use “may be”?
Line 50, “LOX catalytic domain” to “Catalytic domain of LOX”.
Line 55-56, please provide a reference for the conclusion.
Lien 57- 61, please provide a reference for this conclusion.
Line 87-101, please just provide clear description of medium used in this study, but not results and discuss.
Line 94, what dose “modified from” mean?
Line 113, “Purified polynucleotide fragments HTU-AA and HTU-QV”. What does they refer to? Please give a specific explanation or reference.
Line 152 and 154, “TBST” to “TBST buffer”?
Line 179-180, “refolding efficiency is not significantly affected by NaCl concentration (Fig. 2B)”. In Fig. 2B, the activity of the enzyme change from 100% to <70%, which is obviously changed. Please revise this conclusion carefully. In addition, two peaks were showed in Fig. 2B, please reperform and check the results carefully.
Line 186, “in the cold”? please give a clear temperature or temperature range.
Line 187-189, the data should be shown in the manuscript, because this is important results. Or please delete this conclusion.
Line 213, “this band was reliably detected only at a high sensitivity”, how is the sensitivity? And the images in figure 4 are not in good qualities. Please reperform the experiment and get a better picture.
Line 226-227, please give the method and data about this result.
Line 230, more discussions are needed on sequence alignments of Fig. 5-7.
Line 232, “hyperconserved amino acid residues like Cu-binding His and TPQ formation.” Please provide the references for it.
Line 233-235 and Fig 8, please give a clear note of each genes (from which species and their classification in kingdom or phylum).

---

## Round 0.2 · Major Revisions

I agree with the reviewer that the phylogeny portion of this manuscript fits uneasily with the rest, and I honestly cannot see how it improves on the previous analysis of possible HGT of these enzymes. The large number of bands observed in the blots also seems to me (like our reviewer) to show that your antibodies are quite unspecific. How can you be sure that all of your bands really come from LOX? Even if you can present (as suggested) a control showing that your antibodies do bind to purified recombinant LOX, such a control would not be able to show that they ONLY bind to LOX and therefore I cannot see how your speculations regarding possible multimeric forms, etc. are tenable.

Reviewer 2 ·

Basic reporting

Effort has been made in proper citing previous works on the field and in the presentation of experimental results. No more comments about this particular section.

Experimental design

I still feel that manuscript is a complicated mixture of phylogenetic and experimental results that hardly fit together.

Protein methodologies are still no acceptable. As explained below, I do not yet understand why authors do not validate their antibody with the recombinant proteins they have generated. Conclusions raised on these results are fully speculative.

Validity of the findings

With respect to the findings, the authors still claim that they were able to express, purify and refold a couple of deletion mutants of LOX from H. turkmenica, whereas Fig. 1 still shows only the expression of one of them. It is true that supplemental figures shows the two bands in a gel, but there is no information about how bacterial expression of the HTU-AA is induced.
The authors still claim that they are generated a polyclonal antibody against the H. turkmenica protein and used it to characterize the expression of this form in these cells. In fact, quality of the blot is still poor, and it is difficult to believe that the bands now appearing are in fact the proteins they expect. Validation of the antibody using the recombinant proteins is missing and this would be an important control. I feel it is too premature to say that high molecular weight precursors of LOX are expressed in these cells just by looking those heavy bands in the gel. Even more unrealistic is the statement that the proteolytic events are taking place to explain the different bands.

Additional comments

In general, I feel that the manuscript still presents serious concerns regarding the detection of the lox forms expressed in H. turkmenica cells as well as requires further characterization of the generated recombinant proteins.

---

## Round 0.3 · Major Revisions

While I appreciate the improved analysis with the purified antibody, I am sorry to communicate that your manuscript is still not ready for acceptance. Specifically:

- I am not sure that the discrepancies in molecular weight have been reconciled satisfactorily: the text claims that "since LOX is a rather acidic protein this peculiarity should be considered as natural." but neither supporting references or an explanation for this "naturalness" are presented

- I think that reviewer 1 is not convinced that the 210 kDa band does come from LOX;I also cannot be sure of that, especially considering that the purified antibody can still bind non-specifically to a large number of proteins. Incidentally, I think the manuscript text does not explain (in contrast to your rebuttal) why you used the antibody against Halorubrum proteins.

- the phylogenetical analysis (which has problems of its own, as detailed by reviewer #4) still does not seem to provide any new insight to the rest of the paper.

Reviewer 1 ·

Basic reporting

'no comment'

Experimental design

'no comment'

Validity of the findings

'no comment'

Additional comments

The authors tried to show the presence of HTU-LOX in cultured cell. However, there is discrepancy in the molecular weight of proteins. The calculated molecular weight of the recombinant with the His-tag is 24.3kDa but the protein expressed in E. coli was 34 kDa. However, they detected a protein with 50kDa in cultured cell in addition to 210 kDa protein. So, the relationship among these proteins can not be explained. Furthermore, the 50kDa protein was detected with the non-specific antibody as shown Fig. 4.
If the authors expect that the 210kDa protein is a multimer, I wonder why they did withelectrophoresis with but that do not contain mercaptoethanol.
Also, the authors need to show the enzyme activity from host cell extract where they detected the protein by western blot.
Therefore, despite of revision, the questions raised from the first review have not been solved yet

Reviewer 4 ·

Basic reporting

The article is clearly written and its experimental conclusions are well-supported, with some reservations regarding the validity and relevance of their phylogenetic analyses (see below).

Comments on the introduction:

Line 56-58 – The authors claim that “all animal genomes sequenced to date” have LOX genes (with exceptions). The cited study (Grau-Bové et al. Sci Rep 2015) does not support this strong claim. I suggest it is amended to “most animal genomes” or a similar phrase. Incidentally, the placozoan animal Trichoplax adhaerens should also be included among the exceptions (alongside Nematoda and Ctenophora).

Line 62 – The fact that archaeal LOX genes derive from a bacteria-to-archaea HGT is not an established fact, but rather remains a hypothesis. Confirmation of this hypothesis should include at least the identification of a donor bacterial lineage (currently impossible, given the lack of statistical supports in the phylogeny (Grau-Bové et al. Sci Rep 2015)). Thus, the authors should mention HGT as hypothetical (here and elsewhere in the paper), and then state that it is indeed the most parsimonious explanation given current phylogenetic and taxonomic knowledge.

Line 77 – I would suggest that the authors introduced a new paragraph here (“Haloterrigena turkmenica was isolated from Turkmenistani sulfate saline [...]”), for clarity.

Experimental design

See specfic comments about methods for phylogenetic analysis in "Validity of the findings". No further comments

Validity of the findings

Line 303-306 – The authors claim: “Although the second option is less likely due to splitting of animal ORFs into exons, it allows a more parsimonious explanation of the evolution of the catalytic LOX domain both in animals and bacteria”. This is a very strong claim that does not fit neither their results (Fig7 – see problems listed below) nor previous work (Grau-Bové et al 2015 Sci Rep). Animal LOX genes are monophyletic (bacterial sequences do not appear within the animal part of the tree) and actually descend from a LOX homolog that was already present in the last common ancestor of animals, ichthyosporeans and fungi.

Line 306 – LOX HGTs are not “obvious facts”. While they are a likely explanation and this is a valid background for the interest of LOX in Haloterrigena turkmenica, they remain hypothetical (see discussion in the introduction).

Line 262 and Fig. 7 – I do not agree with the authors’ interpretation of their phylogenetic analysis. They claim that LOX genes appeared in actinomycetes and then ‘spread’ to other prokaryotes. This assertion is ambiguous: do they refer to HGT or vertical inheritance? If they were talking about to HGT, how many HGT events do they claim did occur? My interpretation of their tree would not confidently support any (see below). If they meant vertical inheritance, they provide no rationale for that assertion of an actinomycete origin (e.g. some independent information that allows them to root the tree). Most importantly, the tree does not address the specific HGT event that is hypothesized to have transferred LOX to the ancestor of Haloterrigena turkmenica.

In addition to these questions of interpretation, there were major methodological issues which I will list below.

However, the phylogenetic analyses here performed are not essential to interpret the valuable functional analyses detailed in the results section. Thus, given that it does not affect the validity of the rest of the work, I would recommend that this analysis and discussion be removed from the present manuscript before publication. Whenever needed, the authors can refer to the results from previously published analyses (Figure S1 and S2 from (Grau-Bové et al. Sci Rep 2015)).

As an alternative, the authors could address all the methodological concerns (but this would require exhaustive reanalysis and rewriting – I leave this choice for the consideration of the editor and authors).

These are the methodological issues:

(i) The aim of the phylogenetic analysis is not clear. As an example, it would have been interesting to identify the possible donor for the bacteria-to-archaea HGT. However, this issue is not addressed in neither the discussion nor the results. As a specific example, they could include a wider taxon sampling to identify close bacterial relatives of archaeal LOX that could be candidate donors (it would require some improvement in the taxon sampling compared to previous analysis, which is not the case – cf Figure S1 and S2 from (Grau-Bové et al. Sci Rep 2015).

(ii) The authors claim to have used approximate likelihood ratio test as statistical supports for their nodes, but this information is not displayed in Fig. 7. The authors should always display the statistical supports in all nodes of the tree. In addition, the interpretation of the aLRT metric in isolation can be unintuitive (see for example Minh et al MBE 2013, PMID: 23418397), and it is good practice to report more than one method of statistical support for the ML phylogeny (e.g. standard bootstrap replicates or UFBS). For example, Fig7 shows Sorangium cellulosum as sister group to Haloterrigena turkmenica, but without statistical support it cannot be determined whether the relationship is significant or not.

(iii), the authors should have used a phylogram instead of a cladogram (i.e. where branch lengths are proportional to phylogenetic distance). Following the previous example, this would allow the readers to examine whether Haloterrigena turkmenica is sufficiently close to Sorangium cellulosum to be deemed a good candidate for HGT donor.

(iv) Finally, the authors should include a more detailed description of their phylogenetic methodologies in the main text (Methods section). These details are important and should not be confined to a figure’s caption text. These improved methods must include: 1) how were the sequences retrieved from GenBank (blast? HMMER? From a previous publication? Etc.); 2) how did the authors define the catalytic region they used for the analysis; 3) how were the sequence alignments constructed and how many positions were included; 4) how were the alignments trimmed and how many positions were included; 5) which substitution model did the authors use (LG, WAG, did they include gamma categories, etc.) and how/why did they choose it (model selection can be assessed with dedicated software e.g. ProtTest; or with IQ-TREE, RaxML, etc.); 6) finally, as mentioned above, the authors should include, at least, statistical supports from standard bootstrap replicates (implemented in all major phylogeny inference tools e.g. Phyml, IQ-TREE, RAXML).

Additional comments

I fear that the use of ‘cryptic’ in in the title can be misleading. It suggests a ‘hidden’ role for the enzyme (e.g. in comparison to something else), but in actuality it is simply ‘unknown’ (hence the study). Elsewhere in the article, the correct meaning is clearer.

---

## Round 0.4 · Minor Revisions

Before I send your paper to the reviewers, I must ask you to include in the text an explanation of the use of the Halorubrum proteins. Please also rewrite the discussion to provide full paragraphs instead of bullet points.

---

## Round 0.5 · Minor Revisions

As you see, there are still a few minor points to clarify. I do not expect them to be too problematic.

Reviewer 1 ·

Basic reporting

No more comment

Experimental design

No more comment

Validity of the findings

No more comment

Additional comments

Line 187: Is this true mutants? It is a shorten form of LOX protein and not a real mutated form of an organism.
Line 194: the authors clam that the slow mobility of this protein is because it is acidic protein. Does the removal of N-terminal signal peptide the acidity of this protein? How was the mobility of full length HTU-LOX?
Line 126. Please indicate the location of AA and QV in LOX.
Lines 258-261 can be discussed in the Discussion section
Lines 296-297. What result supports this discussion (Refer the Figure).
The authors reported the assay for enzyme activity, but did not indicated how much enzyme they used for enzyme activity. On line 28 of the abstract, the authors reported that cultured H. turmenica did not have any enzyme activity but it is not described in the material and Methods, and Result section. How this experiment conducted? How much cell extract was used. Althoug they did not detected enzyme activity from the cultured cell, they detected a protein of 210kDa. If there is that much enzyme that present in the cell, why there is no enzyme activity? By running a known amount of purified protein with the cell extract, the authors can tell whether the absence of the activity is because the amount of enzyme in the cultured cell. Is the absence of enzyme activity because of the presence of antifuntional signal peptide on the 210kDa form? This could be discussed..

---

## Round 0.6 · accepted · Accept

I am satisfied with your responses and the changes you made.

#